# Effect of Bleaching Processes on Physicochemical and Functional Properties of Cellulose and Carboxymethyl Cellulose from Young and Mature Coconut Coir

**DOI:** 10.3390/polym15163376

**Published:** 2023-08-11

**Authors:** Warinporn Klunklin, Sasina Hinmo, Parichat Thipchai, Pornchai Rachtanapun

**Affiliations:** 1Division of Marine Product Technology, School of Agro-Industry, Faculty of Agro-Industry, Chiang Mai University, Chiang Mai 50100, Thailand; 2Center of Excellence in Agro Bio-Circular-Green Industry (Agro BCG), Faculty of Agro-Industry, Chiang Mai University, Chiang Mai 50100, Thailand; 3Master of Science Program in Physical Chemistry, Faculty of Science, Chiang Mai University, Chiang Mai 50200, Thailand; hinmo7231@gmail.com; 4Doctor of Philosophy Program in Nanoscience and Nanotechnology (International Program/Interdisciplinary), Faculty of Science, Chiang Mai University, Chiang Mai 50200, Thailand; parichat_thi@cmu.ac.th; 5Division of Packaging Technology, School of Agro-Industry, Faculty of Agro-Industry, Chiang Mai University, Chiang Mai 50100, Thailand

**Keywords:** agricultural waste, coconut coir, cellulose, carboxymethyl cellulose, CMC, bleaching

## Abstract

The objective of this study was to characterize the properties of cellulose and CMC synthesized from young and mature coconut coir with different bleaching times (bleaching for the first time; 1 BT, bleaching for a second time; 2 BT, and bleaching for the third time; 3 BT) using hydrogen peroxide (H_2_O_2_). The surface morphology, structural information, chemical compositions, and crystallinity of both cellulose and CMC were determined. H_2_O_2_ bleaching can support delignification by reducing hemicellulose and lignin, as evidenced by FTIR showing a sharp peak at wave number 1260 cm^−1^. The cellulose and CMC from coconut coir can be more dispersed and have greater functional characteristics with increasing bleaching times due to the change in accessibility of hydroxyl groups in the structure. The CMC diffraction patterns of coconut coir after the bleaching process showed the destruction of the crystalline region of the original cellulose. The SEM images showed that the surface of CMC was smoother than that of cellulose. The CMC_y_ had a higher water holding capacity (WHC) compared to the CMC_m_ as the bleaching can increase interaction between the polymer and water molecules. Therefore, the best quality of CMC corresponds to CMC_y_. Based on these findings, bleaching time has a strong effect on the functional properties of cellulose and CMC from coconut coir.

## 1. Introduction

The coastal regions and islands of Oceania and Asia, such as the Philippines, Indonesia, and Thailand, accounted for 86% of the global coconut cultivation area (12 million hectares), which produced 1.8 million tons of coconut coir and 0.75 million tons of coconut shell remaining due to consumption and discard [1]. The coconut needs to be processed before the fruit is sold. Therefore, it produces a lot of waste material.

Coconut sheaths are in the husk or mesocarp (mesocarp), and most of them are 45.84% lignin and 43.44% cellulose [2], which can be used to generate natural fuel, improve soil physical conditions, and remove industrial pollutants. The use of coconut shells in the various areas mentioned above is of low value. Coconut coir (coconut fiber) can be removed from the husk and is an internationally traded product as a raw material in non-food products due to its quantity, price, and non-toxicity [3]. Coconut coir is high in lignin and cellulose, which makes it stronger than most other types of natural fibers. However, waste generated from coconut processing, when discarded or burned, can create an environmental problem. Many types of research into the use of coconut coir focused on bioethanol [3], catalysts in biodiesel production [4], biodegradable composite films [5], surface mortar reinforcement [6], particleboard production [7], and growing medium for plants [8]. Coconut coir usually has a light brown or tan color and is not commonly used as a source of cellulose due to its natural color. The color of coconut coir is due to its lignin content. Lignin is known to contribute to the brown coloration of various plant materials, including coconut coir. To purify cellulose, lignin and hemicellulose are removed through a bleaching process achieved either via delignification or color-stripping, both of which might damage some cellulose. Breaking the various types of bonds within lignin, different compounds such as hydrogen peroxide (H_2_O_2_) are used as bleaching agents during the bleaching process [9]. This process helps to lighten the color of the fiber and can result in a whiter or more neutral shade. It is important to note that the bleaching process can affect the physical properties of the fiber, including its strength and durability [9]. Various chemical and physical treatments can also be used to modify cellulose from both wood and non-wood sources [10]. One of the most modified celluloses is carboxymethyl cellulose.

CMC is versatile cellulose for several applications, such as for use as a thickening agent [11], binder [12], coating agent [13], film [14], and indicator film [15] or being composited in edible films. CMC is prepared by the conventional slurry process using aqueous sodium hydroxide activation cellulose by reacting with the cellulose and monochloroacetic acid as an etherifying agent. Carboxymethylation occurs when the phase-separated system is converted with monochloroacetic acid or its sodium salt. The hydroxyl groups of the cellulose chains are stimulated and transformed into more reactive alkalines in this process, which is termed alkalization form (CLL−ONa) [16].
CLL−OH + NaOH → CLL−ONa + H_2_O (1)

Following this, an etherification is performed as in Equation (2) to obtain CMC, and a side reaction is then performed as in Equation (3), which results in sodium glycolate [11].
CLL−ONa + Cl−CH_2_−CO−ONa → CLL−O−CH_2_−COONa + NaCl (2)
NaOH + Cl−CH_2_−CO−ONa → HO−CH_2_−CO−ONa + NaCl (3)

The demand for CMC mainly driven by the increasing demand for ready-to-eat and low-fat food is positively impacting the CMC market’s growth. The market trend for CMC is expected to rise from approximately USD 1.67 billion in 2021 to USD 2.23 billion in 2028, with a compound annual growth rate of 4% between 2023–2028 [17,18,19]. CMC, also known as cellulose gum, can be extracted from non-wood sources; for example, orange peel [20], rice straw [21], durian rind [16,22], corn husk [23], corn peel [24], asparagus stalk end [25], papaya peel [12,26], palm bunch [27], banana tree [28], mimosa *pigra* peel [29], mulberry paper, and bacterial cellulose [30].

No research has presented the study of CMC powder synthesized from young coconut coir (CMC_y_) and mature coconut coir (CMC_m_) so far. Coconut coir is not commonly used for CMC production for several reasons: (1) Coconut coir contains a relatively high amount of lignin compared to other cellulose sources [2]. Coconut coir is a heterogeneous material with variations in fiber length, lignin content, and cellulose crystallinity across different husk parts. This non-uniform composition can result in inconsistent reaction kinetics and product quality during carboxymethylation [11]. (2) Coconut coir fibers have high tensile strength and are difficult to separate and disintegrate into individual cellulose fibers. Effective pretreatment methods are required to remove impurities, lignin, and hemicellulose from the coconut coir before the carboxymethylation process [5]. These additional processing steps can increase the overall cost and complexity of CMC production from coconut coir. The challenges of synthesizing CMC from coconut coir were associated with high lignin content, non-uniform composition, processing difficulties, and cost considerations. Therefore, this study aimed to determine the effect of the bleaching process using hydrogen peroxide (H_2_O_2_) on the degree of substitution (DS), percent yield, lignin content, viscosity, color values, water, oil absorption capacity, water and oil holding capacity, water and oil retention capacity, water solubility, functional groups, and morphology of both cellulose and CMC from young and mature coconut coir.

## 2. Materials and Methods

### 2.1. Materials

Young and mature coconut coir were purchased from Chang Moi Sub-District, Chiang Mai, Thailand. All chemicals used in the preparation, synthesis, and analysis of CMC were AR grade or equivalent. Glacial acetic acid and sodium hydroxide were from Lab-scan, Chiang Mai, Thailand, hydrogen peroxide (10%) from QReCTM (Merck & Co., Inc., Darmstadt, Germany), isopropyl alcohol (IPA, Monochloroacetic acid) from Sigma-Aldrich (Sigma-Aldrich, Steinheim, Germany), absolute methanol, ethanol, and sodium chlorite from Union Science Co., Ltd. (RCI Labscan Ltd., Bangkok, Thailand).)

### 2.2. Materials Preparation

Both young and mature coconut coir were peeled and then cut into small pieces, size 6 × 10 cm, before washing with 500 ppm chlorine solution for 5 min. The slurry of young and mature coconut coir was dried in a hot-air oven (Memmert, Büchenbach, Germany) at 105 °C for 6 h. Dried coconut coir was ground by using a grinder (ML-SC5-III, Min Lee Industrial Ltd., Shanghai, China) before being crushed with the high-speed blender (DXM-700-F, Dxfill machine, Guangzhou, China) at a speed of 35,000 rpm for 5 min until the particle size of dried coconut coir powders was smaller than 1 mm. The dried powder samples were stored in polypropylene (PE) bags at room temperature until use. These processes were slightly adapted from a previous work of Klunklin et al. [25].

### 2.3. Extraction of Cellulose from Young and Mature Coconut Coir

Both young and mature coconut coir were subjected to the alkaline treatment with 15% (*w*/*v*) NaOH solution in a ratio of 1:15 treated at 90 °C for 3 h under a continuous stirrer (IKA RW 20 digital, BEC Thai Bangkok Equipment & Chemical Co., Ltd., Bangkok, Thailand) at a speed of 500 rpm. The slurry was filtered and rinsed with water until the natural pH was reached. The residue was then dried in an oven at 55 °C for 24 h to obtain pulp. From the pulp were removed hemicellulose and lignin by treatment with 1% (*w*/*v*) sodium chlorite (NaClO_2_) and glacial acetic acid in a ratio of 1:10. The pH of the cellulose solutions was controlled by rinsing with distilled water until pH reached around 6.5 to 7. Cellulosic pulp fiber underwent three different bleaching times (1 BT, 2 BT, and 3 BT) of 10% (*w*/*v*) hydrogen peroxide (H_2_O_2_) bleaching. The fiber residue from celllose_y_ and cellulose_m_ was dried again in the oven at 55 °C for 24 h before being stored in plastic bags at room temperature for further use [10]. The cellulose_y_ and cellulose_m_ products were in powder form after synthesis. The yield, expressed as a percentage, was calculated using Equation (4):(4)$$Yield\;of\;cellulose\left(\%\right)=\frac{Weight\;of\;cellulose\left(g\right)}{Weight\;of\;powder\;coconut\left(g\right)}\times 100$$

### 2.4. Carboxymethyl Cellulose (CMC) Synthesized from Young and Mature Coconut Coir

CMC_y_ and CMC_m_ from cellulose_y_ and cellulose_m_ were both synthesized according to the method described by Rachtanapun et al. [16] and Klunklin et al. [25]. Fifteen grams of cellulose_y_ and cellulose_m_ powder were mixed with 50 mL of 30% NaOH and isopropanol (450 mL) and soaked for 30 min at ambient temperature before adding 18 g of monochloroacetic acid. The mixture was separated into two phases by standing for 3 h. The solid phase (70 mL) was suspended in 100 mL methanol and neutralized with glacial acetic acid. Then, the suspension was filtered and washed with 70% ethanol 5 times before rinsing with absolute methanol and being filtered again. The residue was dried at 55 °C in a hot-air oven overnight for obtaining CMC_y_ and CMC_m_. The percent yield was calculated by the following Equation (5).
(5)$$Yield\;of\;CMC\left(\%\right)=\frac{Weight\;of\;CMC\left(g\right)}{Weight\;of\;cellulose\left(g\right)}\times 100$$

### 2.5. The Determination of Lignin Content

Lignin content was determined using the TAPPI method by following the procedure outlined in T222-om-88 [31]. A sample of 1 g was digested in 15 mL of 72% H_2_SO_4_ for 2 h. The sample was autoclaved at 121 °C for 60 min before cooling and filtering with filter No. 4. Then, the sludge was treated with hot water 250 mL and dried at 105 °C for 6 h. Lignin can be calculated using the following Equation (6) by Rachtanapun et al. [9].
(6)$$The\;percentage\;of\;lignin=\frac{\left(Weight\;of\;residue\;content\right)}{Weight\;of\;CMC\;content}\times 100$$

### 2.6. The Kappa Number of Pulp

Testing of the cellulose fiber was carried out to determine the bleaching ability or degree of delignification of the coconut coir based on the standard method of TAPPI-T236 om-99 as a Kappa number [32]. The Kappa number is the amount (in mL) of 0.1 N potassium permanganate solution consumed by one gram of moisture-free pulp under the conditions specified in the method. Briefly, the coir was suspended in distilled water before adding 0.1 M potassium permanganate and 4 M sulfuric acid under continuous stirring for 5 min. The cessation reaction was carried out by using 1 M potassium iodide. The mixture was then titrated immediately with 0.2 M sodium thiosulfate [9]. The kappa number can be calculated using the following Equation (7).
(7)$$p=\frac{\left(b-a\right)N}{0.1}\;and\;K=\frac{pf}{W}$$
where K = Kappa number, f = correction factor (g/mL), w = weight of moisture-free pulp (g), p = amount of 0.1 N permanganate consumed (mL), b = amount of the thiosulfate consumed in the blank (mL), a = amount of the thiosulfate consumed by the test specimen (mL), N = normality of the thiosulfate (N).

### 2.7. Color Characteristics of Cellulose and CMC Samples

The color characteristics of all cellulose and CMC samples were evaluated by using a Color Quest XE Spectro colorimeter [25] (Hunter Lab Colorflex EZ 45-0 (LAV), Shen Zhen Wave Optoelectronics Technology Co., Ltd., Xiamen, China) to measure the CIELAB system as three color parameters: L*, a* and b* where L* represents the lightness from black (0) to white (100), a* represents the position between greenness (−) to redness (+), and b* represents the yellowness from blue (−) to yellow (+). The total chromatic variations (∆E) were calculated from L*, a*, and b* values of cellulose and CMC powders and calculated using Equation (8).
(8)$$\Delta E\;=\sqrt{{\left(\Delta L^{*}\right)}^{2}+{\left(\Delta a^{*}\right)}^{2}+{\left(\Delta b^{*}\right)}^{2}}$$

### 2.8. Fourier Transform Infrared Spectroscopy (FTIR)

The functional groups of all cellulose and CMC samples were analyzed using FTIR spectroscopy (Tensor 27, Bruker Optik GmbH, Ettlingen, Germany). Dry samples were ground and pressed into a pellet mixed with powdered KBr. The FTIR spectra were recorded in the wavenumber range of 4000–400 cm^−1^ [33].

### 2.9. X-ray Diffraction (XRD) of Cellulose and CMC Samples

Samples were dried in a hot-air oven (Memmert, Munich, Germany) at 105 °C for 3 h before testing. X-ray diffraction patterns were obtained from powdered samples carried out by an X-ray powder diffractometer (JEOL, JDX-80-30, Shimadzu, Kyoto, Japan) in the reflection mode on a JEOL with a scan speed of scattering angle (2θ) at 5°/min in the scan range from 10 to 60° [26].

### 2.10. Scanning Electron Microscopy (SEM) of Cellulose and CMC Samples

The surface morphology of all cellulose and CMC samples were analyzed by using SEM (LV-SEM: JSM 5910 LV, JEOL Ltd., Tokyo, Japan). The prepared samples were mounted onto a sample stub and sputter-coated with gold before being placed in a scanning electron microscope (SEM). The samples were analyzed through a large field detector. The acceleration voltage was used at 15 kV with 500× and 1500× original magnification.

### 2.11. Determination of the Degree of Substitution (DS) of CMC_y_ and CMC_m_

The DS of CMC_y_ and CMC_m_ were determined using the USP XXXII method standard for croscarmellose sodium as described by Rachatanapun and Rattanapanone [29].

### 2.12. Water/Oil Absorption Capacity

The water/oil absorption capacity of cellulose and CMC powder samples was determined based on the method as reported in the literature [22]. The water/oil absorption capacity was calculated based on Equation (9) [34,35].
(9)$$Water/Oil\;absorption\;capacity=\frac{\left(W-W_{0}\right)}{\left(W_{0}\right)}$$
where W is the weight of cellulose and CMC powder samples after the water/oil absorption test and W_0_ is the initial weight of cellulose and CMC powder samples.

### 2.13. Water/Oil Holding Capacity

Sample 0.5 g was added to 25 mL of distilled water/vegetable oil. A tube was placed in a forced-air oven at 37 °C overnight with moderate shaking. After that, the tubes were centrifuged for 10 min at 2000 rpm. The surplus water was decanted, and the tubes were inverted and left to drain for 10 min. Each tube was weighed separately (W_2_). By deducting the weight before water treatment (W_1_), the amount of water stored was computed and expressed as dry weight calculated based on Equation (10) [36].
(10)$$Water/Oil\;holding\;capacity=\frac{\left(W_{2}-W_{1}\right)}{W_{1}}$$
where W_1_ is the initial weight of cellulose and CMC samples and W_2_ is the weight of cellulose and CMC samples after the determination.

### 2.14. Water/Oil Retention Capacity

One gram of samples was put into test tubes and weighed (W_1_). Then, 10 mL of distilled water was added to each sample before mixing for 1 min. The samples were allowed to stand for 30 min at 25 °C before being centrifuged at 4000× *g* for 15 min and then drained by tilting the test tubes 10 degrees from horizontal for 10 min. The weight of the test tubes (W_2_) was determined, and the amount of water retained was computed as follows in Equation (11) [37].
(11)$$Water/Oil\;retention\;capacity=\frac{\left(W_{2}-W_{1}\right)}{W_{1}}$$
where W_1_ is the initial weight of cellulose and CMC samples and W_2_ is the weight of cellulose and CMC samples after the determination.

### 2.15. Water Solubility

Samples were soaked in distilled water (50 mL) and shaken continuously for 2 h. The residue was filtered and dried in an oven at 105 °C for 24 h. The samples were calculated using Equation (12) [38].
(12)$$Water\;solubility=\frac{\left(W_{2}-W_{1}\right)}{W_{1}}$$
where W_1_ is the weight of cellulose and CMC samples after the dry weight of water leached and W_2_ is the initial weight of cellulose and CMC samples.

### 2.16. Viscosity of CMC Samples

Sample solutions (1 g of each sample/25 mL of water) were mixed by using continuous stirring (IKA RW 20 digital, BEC Thai Bangkok Equipment & Chemical Co., Ltd., Bangkok, Thailand) with a stirring speed of 500 rpm for 1 h at room temperature. The viscosity measurement was carried out at the speed of 200 rpm for 10 s at 25 °C a with Brookfield viscometer (LVDV-2, Newport Scientific, Middleboro, MA, USA) [39].

## 3. Results and Discussion

### 3.1. Percent Yield of Cellulose and CMC Samples

The percent yields of cellulose and CMC samples are shown in Figure 1. The bleaching times affected the yield of both cellulose and CMC samples. After bleaching, the yields of cellulose_y_ and cellulose_m_ decreased from 50.34% to 40.76% and 60.76% to 45.78%, respectively. The percent yield of cellulose_y_ showed the same trend compared to cellulose_m_ after bleaching; however, the percent yields of cellulose_m_ were significantly higher (*p* < 0.05) than cellulose_y_ in each treatment. This may be because the higher lignin content and Kappa number were found in cellulose_m_ compared to cellulose_y_, as reported in Table 1. Ranking based on the percent yield of cellulose was cellulose at 1 BT > cellulose at 2 BT > cellulose at 3 BT, which means that the bleaching process can potentially reduce the percent yield of the cellulose [9]. The bleaching process involves the removal of lignin and other impurities from the pulp, resulting in the loss of some cellulose fibers along with the lignin. The removal of fibers reduces the overall yield of cellulose [9,40].

The percent yield of both CMCs from coconut coir was increased after the bleaching process. The highest CMC yield was observed at CMC_m_ at 3 BT (80.43%). In this study, the yields of both CMC_y_ and CMC_m_ were lower than the yield of CMC from coconut fiber (119.3%) reported by Huang et al. [37]. The percent yield of CMC from corn husks was provided with the maximum product yield of 179.04% [41]. A bleaching process can remove lignin and other impurities. By removing those barriers, the cellulose becomes more accessible for carboxymethylation, leading to a higher yield of CMC during the subsequent processing steps [42]. The bleaching process can alter the properties of cellulose fibers by enhancing the reactive towards the carboxymethylation reaction. The bleaching process can introduce functional groups or modify the cellulose structure, making it more amenable to carboxymethylation. Increasing reactivity can result in a higher yield of CMC. However, the purity and quality of the starting material can significantly impact the yield [43].

### 3.2. Lignin Content and Kappa Number of Cellulose

Lignin content and kappa number of celluloses from coconut coir are shown in Table 1. Bleaching time affects both the lignin content and the kappa number of samples. The lowest lignin content in cellulose from coconut coir was discovered after bleaching three times, from which it can be stated that the amount of lignin and the kappa number are barely connected [16]. The lignin content and kappa number of the bleached cellulose showed a similar trend with percent yield (Table 1). These results were related to the reported results by Suriyatem et al. [9]. As part of the chemical bleaching process, the oxidation reaction forms a carboxyl group in lignin, which is ionized in alkaline conditions and increases its solubility. Using the principle of the bleaching process cannot remove all lignin contents from the fiber. Lignin can act as a physical barrier, limiting the accessibility of the cellulose chains to the carboxymethylating agents, reducing the efficiency of carboxymethylation and leading to lower yields of CMC [44].

The Kappa number is directly related to the coir’s residual lignin content, which is related to the coconut coir quality and bleaching ability [9]. Kappa’s number of cellulose samples varied from 14.11 ± 0.11 to 21.23 ± 0.10% (Table 1). The delignification to a lower kappa number is a factor in achieving high brightness in the coir (Table 2). A constant decrease in the kappa number value after each stage was observed. The highest decrease in kappa number occurred after chemical bleaching three times (3 BT). Ferdous, Quaiyyum, and Jahan [42] also reported that the kappa number of rice straw pulp was lower after delignification with chlorine dioxide. This could be related to the acid bleaching of the lignin as described by Ferdous et al. [42].

### 3.3. Appearance and Color of Cellulose and CMC Samples

The appearance of cellulose and CMC from coconut coir at the different bleaching times is shown in Figure 2. Cellulose from coconut coir typically has a fibrous, rough surface texture. CMC from coconut coir has a granular or powdery texture with a light brown or beige color compared to the commercial one [6].

A color measurement was conducted to determine the colors resulting from carboxymethylation reactions. Chemophore compounds are removed during chemical bleaching to improve the cellulose’s brightness, a key attribute for cellulose derivative production. Therefore, the bleaching process removes colored compounds of carboxymethylated lignin from coconut coir. The color values of both cellulose and CMC are shown in Table 2 and Table 3, respectively. The color parameters of cellulose and CMC were significantly changed (*p* < 0.05) due to the various bleaching times. As bleaching times increased, the L* values of cellulose_y_ and cellulose_m_ increased. Cellulose_m_ at 1 BT has a higher total color difference (ΔE) compared to cellulose_y_ (Table 2). After bleaching at 3 BT, cellulose_y_ had a brighter or whiter appearance than cellulose_m_. Mature coconut coir typically contains more lignin, which imparts a brownish or yellowish color to the cellulose fibers. Therefore, cellulose_m_ has a dark coloration compared to cellulose_y_, which contains less lignin and is often whiter or lighter in color.

The color of CMC was altered from sandy to cream white compared with native cellulose. The L* values of both CMC_y_ and CMC_m_ were also increased after bleaching three times. The bleaching time and carboxymethylation reaction might be responsible for the color change. The ΔE (total color difference) values of both CMC_y_ and CMC_m_ had a similar trend to a* and b* values, with a similar result to Rachtanapun et al. [16]. The WI (whiteness index) of cellulose and CMC samples had similar trends to the L* value.

### 3.4. Fourier Transform Infrared Spectroscopy (FTIR) of Cellulose and CMC Samples

FTIR was used to verify qualitative functional groups. FTIR analyses of both cellulose and CMC of young and mature coconut coir treated with various bleaching times are shown in Figure 3a,b, respectively. The crests of retention are related to the vibration recurrence of the chemical bonds within the cellulose and CMC compounds. One prominent absorption band is observed in the region of 3350–3300 cm^−1^, which corresponds to the stretching vibrations of the hydroxyl (−OH) groups present in cellulose. This peak is attributed to the intermolecular hydrogen bonding network within the cellulose chains. Another notable absorption peak appears around 2894 cm^−1^, corresponding to the stretching vibrations of the C−H bonds in the cellulose backbone [45]. Cellulose and CMC from coconut coir had similar functional groups with the same absorption bands, such as hydroxyl group (−OH stretching) at 3350–3300 cm^−1^, C−H stretching vibration at 2894 cm^−1^, carbonyl group (C=O stretching) at 1589–1586 cm^−1^, hydrocarbon groups (−CH_2_ scissoring) at 1450–1420 cm^−1^, and ether groups (−O− stretching) at 1050 cm^−1^. The intensity of the carbonyl group (C=O stretching), hydrocarbon groups (−CH_2_ scissoring), and ether groups (−O− stretching) in the CMC samples was increased compared to the cellulose samples [46].

Lignin, a complex aromatic polymer found in plant cell walls, exhibits distinct absorption peaks in the IR spectrum. These include strong bands around 1600–1500 cm^−1^ (aromatic ring stretching vibrations) and 1330–1260 cm^−1^ (C−O−C stretching vibrations) [47]. The presence of a sharp peak at a wave number of 1321 cm^−1^ associated with the reduction in hemicellulose and lignin content due to the bleaching process has been effective in removing hemicellulose and lignin from coconut coir. This peak is commonly attributed to the C−O−C asymmetric stretching vibration of hemicellulose, which includes the glycosidic linkages between sugar units in the hemicellulose structure. The absence of lignin vibration in the FTIR spectrum could indicate the successful removal or degradation of lignin from the material being analyzed. The reduction in hemicellulose and lignin content leads to the exposure of more cellulose fibers, resulting in increased brightness and improved pulp quality [48]. On the other hand, the presence of lignin vibrations would suggest that lignin is still present in the sample, indicating incomplete removal or preservation of lignin in the material as shown in Table 1. Impurities such as lignin and hemicellulose can interfere with the carboxymethylation process, affecting the efficiency of CMC synthesis. As a result, the IR spectra of all CMC samples synthesized indicate the typical absorptions of the cellulose backbone and the carboxymethyl ether group consecutively. The signal at around 1450–1420 cm^−1^ is attributed to the asymmetric stretching vibration of carboxylate groups introduced during carboxymethylation [47]. This result revealed that carboxymethylation on cellulose molecules had been replaced [11] to become carboxymethyl cellulose (CMC).

### 3.5. X-ray Diffraction (XRD) of Cellulose and CMC Samples

XRD is an effective method to investigate the crystallinity of cellulose and CMC samples. X-ray diffractograms of cellulose produced from coconut coir and the corresponding carboxymethyl cellulose were recorded as shown in Figure 4a,b, respectively. The degree of crystallinity in the polymer was assessed qualitatively and quantitatively. The cellulose diffraction pattern shows three sharp cellulose peaks at 2θ = 18°, 34°, and 22°. The positions of the cellulose peaks can vary depending on the specific type of cellulose and the degree of crystallinity. In nature, cellulose is found in the semi-crystalline form [2]. The presence of crystalline regions provides cellulose with its strength, while the amorphous regions allow for water absorption and other functional properties [34].

The CMC diffraction patterns show the destruction of the crystalline structure of the original cellulose (Figure 4a,b). There is a weak diffraction pattern at 2θ = 17°, a complete disappearance of the sharp peak at 2θ = 34°, and a replacement of the broad-spectrum peak. As a result, the crystalline peaks of CMC are characteristic of the diffraction pattern characteristic of cellulose [49]. According to XRD theory, the crystallinity of CMC_y_ and CMC_m_ decreased because of bleaching time effects on cellulose structure. This theory suggests that very tiny, imperfect crystals lead to enlarged diffraction [50]. Moreover, the crystallinity of CMC can be relevant to the synthesis method of cellulose before CMC synthesis [51]. All the characteristic peaks of native cellulose have almost disappeared and have become an amorphous phase [26] affecting the water absorption, water retention, and water solubility of CMC samples.

### 3.6. Morphology of Cellulose and CMC Samples

The morphology of coconut coir was characterized by SEM with an acceleration voltage of 15 kV under various magnifications for the powder and pulp of young (Figure 5) and mature coconut coir (Figure 6). The cellulose and CMC of young and mature coconut coir after being treated with different bleaching times were evaluated in Figure 7. Untreated young coconut coir exhibits a fibrous and thread-like appearance (Figure 5a,b). Moreover, coconut coir may contain other components, such as pith or parenchyma cells. These cellular structures may appear as irregular shapes or clusters within the powder, contributing to the overall porous nature of the material. After being treated with NaOH, several changes can be observed compared to untreated coconut coir. The NaOH treatment modified the structure and surface characteristics of the coir fibers. The impurities and lignin from the coir fibers were partially removed, resulting in cleaner and more defined fibers (Figure 5c,d). Mature coconut coir typically consists of long and thick fibers (Figure 6a,b). NaOH treatment leads to a smoother surface texture by dissolving or degrading the outer layers of the fibers (Figure 6c,d). Mature coconut coir is more tightly intertwined and entangled, forming a denser network (Figure 6b), while the young one may be less densely packed (Figure 5b). Fibers in mature coir are more fully developed and tend to have a higher degree of lignification, resulting in a tougher and more rigid structure compared to the young coir. Therefore, mature coconut coir exhibits the presence of some lignin and other impurities, while young coconut coir fibers have a more uniform shape and exhibit a smoother surface after being treated with NaOH.

The SEM images of cellulose_y_ present a similar shape distribution compared to cellulose_m_ (Figure 7). Significant changes on the surface of cellulose_y_ after being treated with bleaching chemical at 1 BT are shown in Figure 7a,1. The initial bleaching treatment removes impurities, lignin, and non-cellulosic materials from the surface of the cellulose fibers, resulting in a cleaner and smoother appearance compared to untreated fibers. There are no structural alterations to the cellulose fibers. Figure 7b,2 show SEM images of cellulose_y_ after being treated with bleaching chemical at 2 BT. Cellulose_y_ fibers at 2 BT tend to separate further from each other and become more distinct and visible. Slight changes in fiber morphology, such as the length and diameter of cellulose_y_ at 2 BT can be observed. Figure 7c,3 show SEM images of cellulose_y_ at 3 BT. The possibility of some fiber degradation after being treated with repeated bleaching processes can be observed as thinner or shorter fibers compared to cellulose_y_ at 1 BT. Therefore, the bleaching process affected changes in the microstructure of the coconut coir cellulose. SEM images of cellulose_m_ at 1 BT, 2 BT, and 3 BT are shown in Figure 7g–i,7–9, respectively. Some impurities and lignin from cellulose_m_ were removed from the first bleaching time (Figure 7g,7). After the second time bleaching (Figure 7h,8), the fiber surfaces become cleaner, and more uniform compared to cellulose_m_ at 1 BT. Bleaching helps to homogenize the cellulose fibers by removing structure variations. As a result, the fibers exhibit improved uniformity in terms of surface characteristics (Figure 7i,9). The fiber surfaces of cellulose_y_ and cellulose_m_ are quite similar after being treated with a bleaching process. However, cellulose_m_ seems to have a longer shape with some damaged fiber surface after being treated with bleaching three times.

The morphology of CMC_y_ and CMC_m_ treated with different bleaching times is also presented in Figure 7. In general, CMC is produced by cellulose alkalization with a carboxymethylation process using sodium monochloroacetate (NaMCA) as a reagent. Figure 7d–f,4-6, are the morphology of CMC_y_ at 1 BT, 2 BT, and 3 BT, respectively. Cellulose typically displays a fibrous and layered structure, while CMC exhibits a more granular and amorphous morphology. The granules of CMC_y_ are generally smaller and less organized compared to the fibrous structure of cellulose. Figure 7d,4 show larger granules of CMC_y_ than Figure 7e,f. By increasing the bleaching times of CMC, the uniformity of the surfaces of CMC_y_ was decreased. The morphology of CMC_m_ at 1 BT, 2 BT, and 3 BT is shown in Figure 7(j–l,10-12), respectively. CMC_m_ surfaces were rougher and more irregular, with a less defined layered structure than cellulose_m_ surfaces. Figure 7j,k exhibit some long fibrous surfaces; however, Figure 7l shows more crystal structure. CMC_y_ displays a rougher surface with irregularities or protrusions due to a less developed and less compact fiber structure than CMC_m_. Moreover, CMC_y_ shows a more homogeneous and consistent morphology due to the relatively uniform nature of the fibers. The surface morphology of CMC is very rough, and the dispersion of microscopic particles was obtained throughout the images. Due to hydrogen bonding between the carboxyl and hydroxyl functional groups, it has a rough and uneven shape [50]. Both CMC_y_ and CMC_m_ molecules show crystal features that are similar to other images reported for the typical molecular structure of CMC [44]. Moreover, a similar observation has also been reported in the literature that the fiber diameter was changed when bleaching chemical concentration was applied [40].

### 3.7. Degree of Substitution (DS) of CMC_y_ and CMC_m_ with Different Bleaching Times

DS refers to the average number of carboxymethyl groups per anhydroglucose unit (AGU) of the cellulose chain and is an important parameter that determines the properties and functionality of CMC. The effect of a DS value of CMC from coconut coir with different bleaching times is presented in Figure 8. The DS value was used to indicate the solubility of CMC. The DS value of commercial CMC ranged from 0.5 to 1.5 [11]. In addition, the DS value of CMC depends on reaction conditions during carboxymethylation, the concentration of reagents, and the reaction time [9]. In this study, the DS values of CMC_y_ and CMC_m_ varied from 0.75 to 0.81 and 0.72 to 0.80, respectively. A high-quality or -performance CMC grade normally has high purity (>90%) with high DS (>0.88) [51]. The presence of lignin in both cellulose_y_ and cellulose_m_ can interfere with the carboxymethylation reaction, reducing the extent of substitution. The cellulose might not properly be dispersed or dissolved in the reaction medium, and the accessibility of the cellulose to the carboxymethylating agents may be limited, leading to a lower DS [41]. The lower DS may affect the functional properties of CMC. In addition, the samples have a high lignin content (Table 1); it is possible that during the carboxymethylation process, both cellulose and lignin can undergo carboxymethylation. The DS values reported in this study might not be specific to carboxymethylated cellulose alone.

Increasing bleaching times have raised the DS value. As the DS value increases, more carboxymethyl groups are introduced, leading to a higher density of carboxylate groups along the cellulose chains. This increased density of carboxylate groups contributes to a more uniform and homogeneous distribution [51]. The carboxylate groups are more evenly distributed along the polymer chains, resulting in more consistent chemical reactivity, solubility, and functionality throughout the CMC structure [27]. The bleaching process using hydrogen peroxide may increase cellulose availability after bleaching [51]. Similar observations have been made in other research, such as CMC from palm bunch [9] and sugar cane bagasse [22] which delignify with H_2_O_2_.

### 3.8. Functional Properties of Cellulose_y_ and Cellulose_m_

The behavior of cellulose in wet conditions is one of the important characteristics of lignocellulosic materials. Bleaching chemicals can either improve or destroy those properties [22]. All functional properties of cellulose_y_ and cellulose_m_ were significantly increased, except water retention capacity (Table 4). Bleaching time affected the characteristics of cellulose in different maturity stages of coconut fibers. Water absorption in the cellulose structure determines the amount of water absorbed by the structure. The water absorption capacity (WAC) of cellulose from coconut coir ranged from 9.48 ± 0.07 to 11.82 ± 0.14%. The WAC of both cellulose_y_ and cellulose_m_ was significantly increased when bleaching time increased. The WAC of cellulose generally depends on various factors, including the degree of crystallinity, the presence of other substances (e.g., hemicellulose and lignin) in the cellulose matrix, and the accessibility of the hydroxyl groups [52]. Highly crystalline cellulose structures tend to have lower water absorption due to the limited availability of hydroxyl groups for hydrogen bonding [53]. Martinelli et al. [52] claim that mature coconut fiber has more amorphous regions in its cellulose structure than young coconut fiber, which is also confirmed in this study (Figure 3). Therefore, the ability of cellulose_m_ to absorb water is much greater than cellulose_y_ (Table 4). Cellulose, a hydrophilic material, does not have a high inherent oil absorption capacity (OAC). However, it can be modified to improve its oil absorption properties by the bleaching process. By increasing bleaching times, the OAC of cellulose_y_ and cellulose_m_ was significantly increased. The highest OAC was found in cellulose_y_ at 3 BT (28.07 ± 0.02%). The water-holding capacity (WHC) of celluloses from coconut coir was significantly increased (*p* < 0.05), being affected by bleaching times. In the case of cellulose at 3 BT, the hydroxyl functional group can form stronger hydrogen bonds with water molecules than the ether functional group, which could explain why cellulose at 1 BT could not hold water molecules as effectively [54]. The bleaching process can alter the arrangement and accessibility of hydroxyl groups in the cellulose structure [9]. The oil holding capacity (OHC) of cellulose_y_ and cellulose_m_ after bleaching with different cycles showed statistically significant differences (*p* < 0.05) among samples. Cellulose_m_ has a low oil holding capacity compared to cellulose_y_. However, the OHC of both cellulose_y_ and cellulose_m_ significantly increased (*p* < 0.05) after the bleaching time increased (Table 4). The presence of a higher number of fine pores and a larger surface area allows for greater OHC to occur after being treated with chemical treatments [55]. Water retention capacity (WRC) is a measure of the ability of fiber samples to retain water which can observe the increase in fiber length. WRC in this study was increased after being treated with different bleaching times, while the oil retention capacities (ORC) of cellulose_y_ and cellulose_m_ fluctuated. The porous structure of cellulose, which consists of interconnected fibers and a network of hydroxyl groups, allows for the retention of water molecules [53]. The bleaching process might modify the cellulose structure to introduce additional functional groups that further improve the WRC. On the other hand, the bleaching process may affect the ORC of cellulose, which removes some of the surface constituents that contribute to oil absorption or alters the surface properties in a way that changes the oil retention capability of cellulose [55]. The ability of the WRC and ORC of cellulose could be different based on the nature of the raw materials. The water solubility (WS) of both cellulose_y_ and cellulose_m_ also increased after the bleaching process. Cellulose_m_ has a higher WS ranging from 9.93 ± 0.01 to 11.60 ± 0.01% compared to cellulose_y_ after being treated with different bleaching times. The bleaching treatment can include the oxidation or degradation of certain functional groups present in the coconut cellulose structure. Therefore, cellulose becomes more susceptible to hydrolysis and WS [44]. The moisture contents (MC) of both cellulose_y_ and cellulose_m_ were increased due to their higher water properties after bleaching from 1.21 ± 0.05 to 1.48 ± 0.05 and 1.52 ± 0.02 to 1.87 ± 0.06%, respectively. The significantly lower percent yield and MC were found in cellulose from coconut fibers reported by Huang et al. [37].

### 3.9. Functional Properties of CMC_y_ and CMC_m_

Determining the number of active polymers in the substance is important to understand the level of CMC used in various products. All functional properties of CMC from coconut coir were significantly different (*p* < 0.05) after bleaching three times (Table 5). Bleaching times affected all characteristics of CMC by increasing all outputs except WAC and MC. A more homogeneous carboxylate distribution can enhance the performance and properties of CMC in various applications. It can improve its water solubility, swelling behavior, rheological properties, and interaction with other molecules or surfaces [56]. WAC of CMC_y_ and CMC_m_ was decreased after being treated with different bleaching times. CMC_m_ at 1 BT has the highest WAC at 42.93 ± 0.05%. Generally, CMC immersed in water was revealed to have good sorption properties compared to 100% cellulose [51]. However, the bleaching processes can lead to chemical reactions that break down the cellulose backbone of CMC, reducing its overall WAC [56]. The electrostatic repulsion and phenomenon of WAC of CMC are influenced by the presence of anionic carboxylate groups, which play a significant role in its behavior in aqueous solutions [57]. WAC and OAC of CMC from coconut coir were similar to the results from Mondal et al. [22]. The application requires the material to absorb water or oil; a higher-absorption capacity would be desirable. For example, in the production of absorbent materials such as paper towels or diapers, a cellulose source with high water absorption capacity would be preferred [51]. WHC of CMC_y_ and CMC_m_ at 3 BT show the highest WHC at 13.47 ± 0.04 and 12.55 ± 0.15%, respectively. Increasing bleaching times can promote better dispersion of CMC particles in water, allowing for increased interaction between the polymer and water molecules, and resulting in improved WHC [51]. Moreover, hydrophilic hydroxyl groups (-OH) are substituted by carboxymethyl groups during etherification [52]. Thus, prepared CMC at 3 BT had a high WHC due to its hydrophilic nature and the bleaching effect. WHC and OHC of CMC_y_ and CMC_m_ were higher than CMC from snake fruit kernel [52]. The highest OHC was found in CMC_m_ at 3 BT (9.90 ± 0.08%) and the lowest OHC was CMC_y_ at 1 BT (9.07 ± 0.02%). Different bleaching times could induce structural changes in the CMC, potentially increasing its ability to interact with and retain oil. These modifications might involve alterations in the polymer chain arrangement or improved oil–CMC interactions [41]. WRC of CMC is crucial in applications where long-lasting moisture or oil retention is required, such as in hydrogel dressings or cosmetic formulations [55]. CMC_m_ illustrated better WRC than CMC_y_ at each bleaching time. Mature coconut coir typically contains more lignin and other natural compounds than young coconut coir. These components can contribute to the structural integrity and enhance the WRC of CMC. A high WRC implied that the CMC was highly hydrophilic [54]. This is probably due to the different bleaching times which might increase the CMC’s ability to absorb water by making more hydroxyl groups accessible to water. The ORC of both CMC_y_ and CMC_m_ at 3 BT were higher than the ORC of CMC_y_ and CMC_m_ at 1 BT ranging from 1.11 ± 0.07 to 1.17 ± 0.02% and 1.20 ± 0.01 to1.31 ± 0.04%, respectively. The ORC of CMC can be influenced by the DS of CMC. Generally, CMC with a higher DS tends to have a greater ORC [9]. WS of CMC_y_ and CMC_m_ were increased after bleaching time increased. WS of CMC_y_ ranged from 80.11 ± 0.01 to 88.51 ± 0.34% and CMC_m_ ranged from 84.34 ± 0.01 to 89.72 ± 0.01% (Table 5). WS can be affected by the distribution of carboxymethyl groups along polymer chains acting as hydrophilic groups, increasing the DS value and the ability of CMC to immobilize water in the system [9,47]. CMC with high water solubility is commonly used as thickeners, stabilizers, or emulsion stabilizers in various industries, including food, pharmaceuticals, and cosmetics [47]. Generally, a CMC with a high WS has a DS value of more than 0.6. Moisture content (MC) was calculated from all volatile substances at the test temperature; all solids in the sample were accounted for as moisture. MC of CMC_y_ was significantly higher (*p* < 0.05) than CMC_m_ by 64.66%. However, the moisture content obtained from this study was lower than the standard procedure (FAO/WHO Expert Committee 2011), which recommends that the moisture content of CMC should not exceed 12% [57]. In general, the MC of CMC can impact its flowability, dispersibility, and stability [56]. From the results obtained, the CMC_y_ was considered pure because it was within the maximum MC (%) allocated for the CMC standard requirement. However, the functional properties of CMC derived from coconut coir were not as good as those of commercial CMC (Table 5) due to several reasons: (1) Impurities in coconut coir may lead to variations in the properties of the resulting CMC, affecting its functionality [52]. (2) Commercial CMC manufacturers carefully control and optimize the DS to achieve good functional properties. In contrast, coconut coir has a lower percent yield with a low DS value, leading to undesirable performance [30]. (3) Alternative cellulose sources may have natural variations in chemical composition and structure, making it challenging to achieve consistent performance [54].

### 3.10. Effect of Bleaching Times on Viscosity of CMC Samples

The viscosity of CMC is an important parameter for industrial use influenced by many factors, such as CMC content [55], NaOH concentration [16,29], temperature [16,29,56], and H_2_O_2_ concentration [9]. Viscosity is used to determine the flow characteristics of fluids or their resistance to gradual deformation by shear stress in products using different CMC concentrations in processing operations [41]. The carboxymethylation process reduces the degree of polymerization (DP) of cellulose, as some of the glucose units are replaced by carboxymethyl groups. The extent of carboxymethylation and the resulting DS affect the DP of the resulting CMC. The DP of cellulose can have a significant impact on its intrinsic viscosity, which is a measure of the resistance of a polymer solution to flow [37]. As displayed in Figure 9, the viscosity of CMC_y_ was increased from 1 BT to 2 BT and remained stable until 3 BT. On the other hand, the viscosity of CMC_m_ increased after bleaching time increased. The bleaching process might cause the degradation of the cellulose chain, CMC structure, and DS value [7]. However, the bleaching process in CMC_y_ may not significantly impact the viscosity due to the minimal structural changes. Among them, CMC_m_ at 3 BT had the highest DS value and high viscosity. This seems to be influenced by the DS value mentioned earlier, attributed to carboxymethyl groups in CMC [16].

## 4. Conclusions

Cellulose was successfully isolated from young and mature coconut coir through alkali treatment followed by delignification with different bleaching process times using H_2_O_2_. All characteristics of both cellulose and CMC of young and mature coconut coir were changed after being treated with different bleaching times. The percent yield of cellulose_m_ was higher than cellulose_y_ at the same bleaching time. Lignin content and Kappa numbers of both cellulose_y_ and cellulose_m_ were decreased after being treated with different bleaching times. The color parameters of both cellulose_y_ and cellulose_m_ were significantly changed due to the bleaching process and lignin content in nature. The functional properties of cellulose from coconut coir were increased after the bleaching time increased. With increasing bleaching times, the percent yields of CMC_m_ were higher than CMC_y_. The viscosity of CMC is dependent on the DS value. The results showed that the maximum DS value of CMC from coconut coir was 0.81, related to a high-water solubility of CMC from coconut coir at 3 BT. The low moisture contents of CMC were observed with a remarkably limited amount of WRC in this exploratory study. The functional properties of WHC, WRC, and WS of CMC from coconut coir were relatively high after bleaching time increased, which provides strength to products for maintaining shape and moisture migration. Given the importance of safety and compliance with regulations in the food and pharmaceutical industries, it is prudent to conduct toxicological evaluations and seek regulatory approval before using CMC from coconut coir.

## Figures and Tables

**Figure 1 polymers-15-03376-f001:**
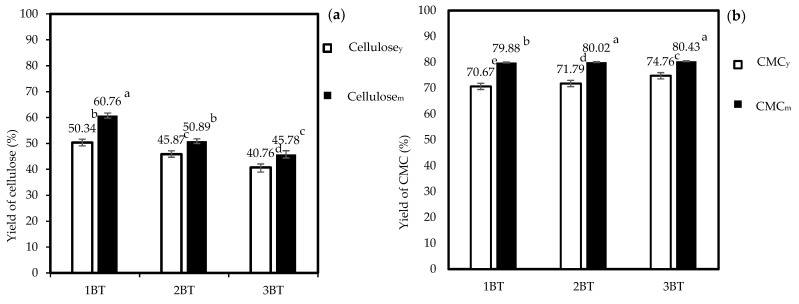
Percent yield of (**a**) cellulose and (**b**) CMC with different bleaching times (1 BT, 2 BT, and 3 BT). Note: values indicated with the same letters are not significantly different at *p* ≤ 0.05.

**Figure 2 polymers-15-03376-f002:**
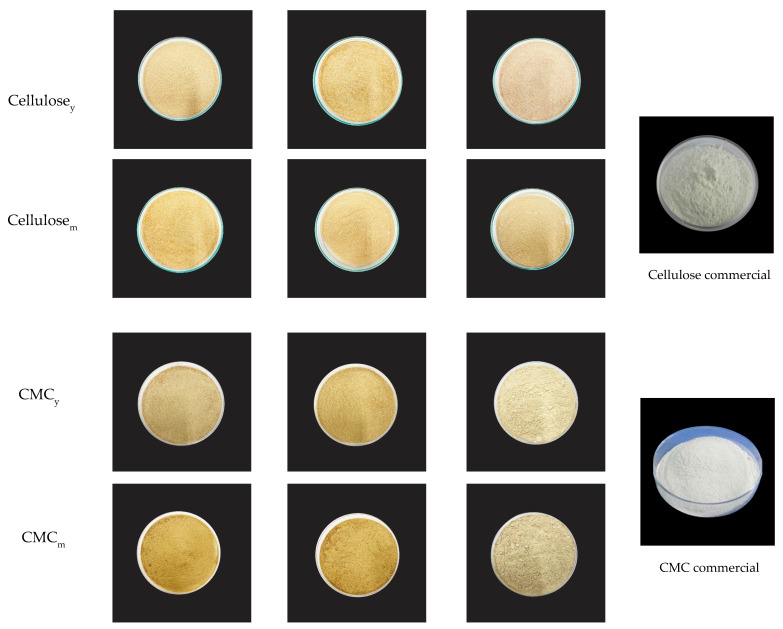
The appearance of cellulose and CMC from young and mature coconut coir at 1 BT, 2 BT, and 3 BT compared to the commercial one.

**Figure 3 polymers-15-03376-f003:**
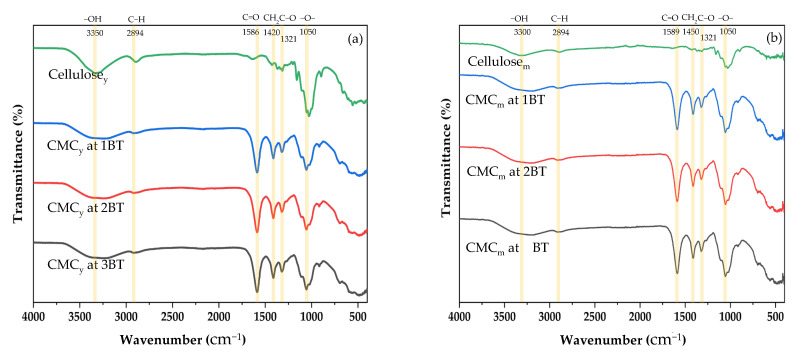
FTIR spectra of cellulose and CMC from (**a**) young and (**b**) mature coconut coir.

**Figure 4 polymers-15-03376-f004:**
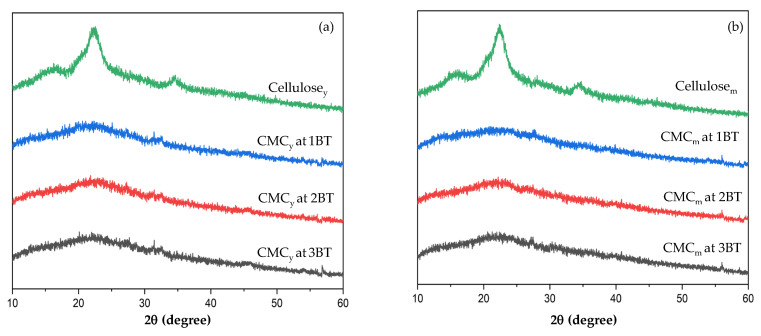
X-ray diffractograms of cellulose and CMC from (**a**) young and (**b**) mature coconut coir.

**Figure 5 polymers-15-03376-f005:**
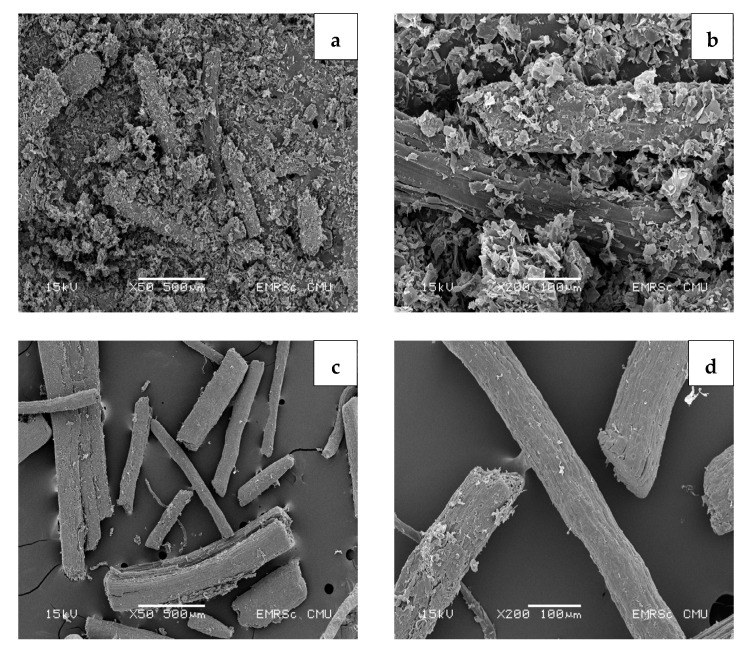
Scanning electron micrographs of (**a**,**b**) young coconut coir, and (**c**,**d**) young coconut coir after being treated with NaOH. The acceleration voltage was 15 kV under two different magnifications at 50× and 100×.

**Figure 6 polymers-15-03376-f006:**
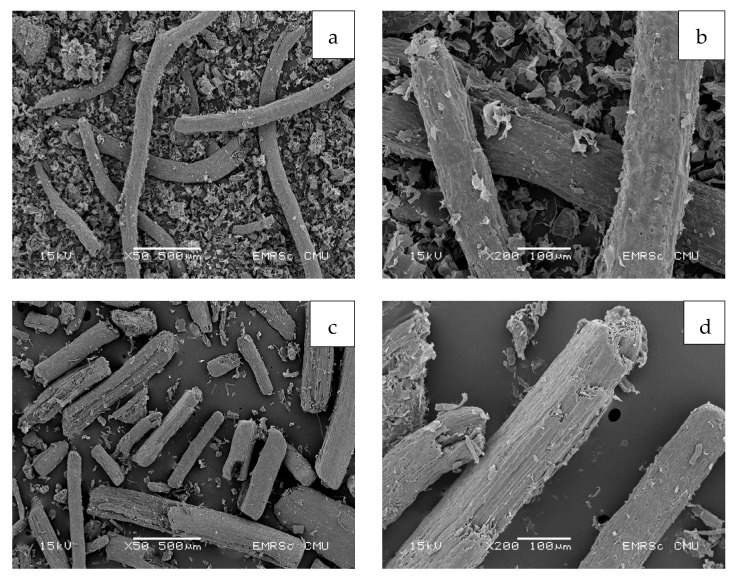
Scanning electron micrographs of (**a**,**b**) mature coconut coir, and (**c**,**d**) mature coconut coir after being treated with NaOH. The acceleration voltage was 15 kV under two different magnifications at 50× and 100×.

**Figure 7 polymers-15-03376-f007:**
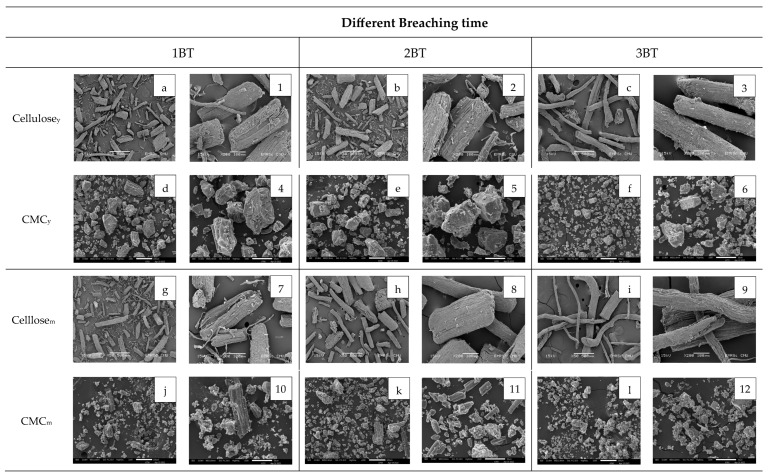
Scanning electron micrographs of (**a**–**c**,**1**–**3**) cellulose_y_, (**d**–**f**,**4**–**6**) CMC_y_, (**g**–**i**,**7**–**9**) cellulose_m_ and, (**j**–**l**,**10**–**12**) CMC_m_ at 1 BT, 2 BT and 3 BT, respectively. The acceleration voltage was 15 kV under two different magnifications, cellulose_y_ and cellulose_m_ at 50× and 200×, CMC_y_ and CMC_m_ at 200× and 500×.

**Figure 8 polymers-15-03376-f008:**
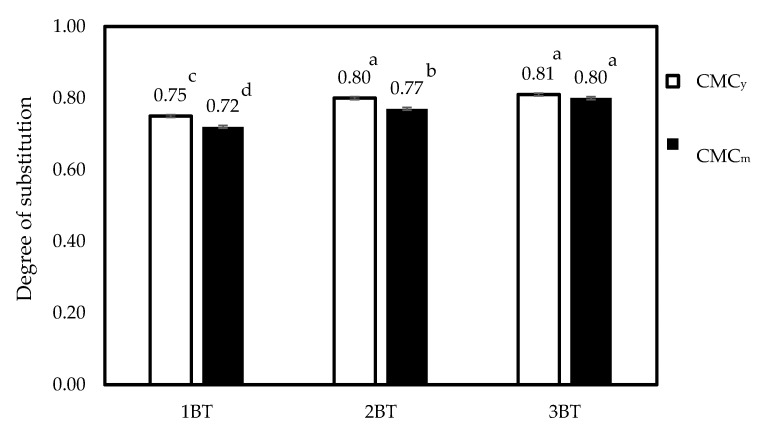
Degree of substitution of CMC_y_ and CMC_m_ with different bleaching times. Note: values indicated with the same letters are not significantly different at *p* ≤ 0.05.

**Figure 9 polymers-15-03376-f009:**
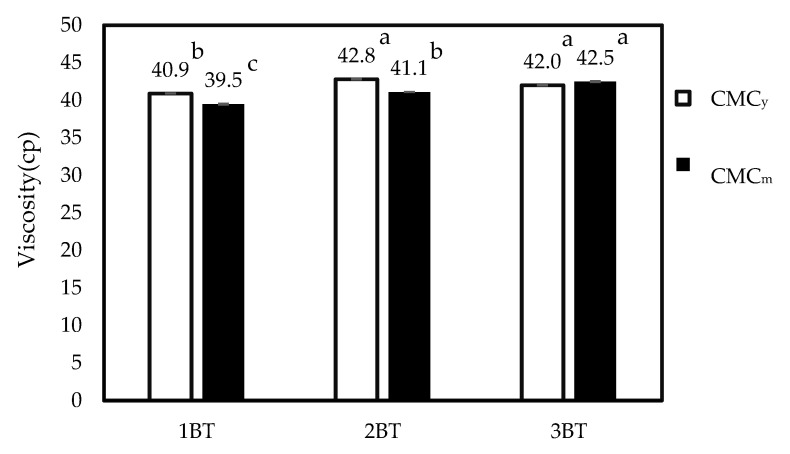
The viscosity of cellulose and CMC samples treated with various bleaching times (1 BT, 2 BT, and 3 BT) Note: values indicated with the same letters are not significantly different at *p* ≤ 0.05.

**Table 1 polymers-15-03376-t001:** Relationship between lignin content and kappa number of Cellulose_y_ and Cellulose_m_ with different bleaching times.

Samples	Bleaching Times	Lignin Content (%)	Kappa Number (%)
Cellulose_y_	1 BT	18.35 ± 0.28 ^c^	20.89 ± 0.14 ^b^
	2 BT	12.21 ± 0.06 ^d^	17.67 ± 0.13 ^d^
	3 BT	11.19 ± 0.28 ^e^	14.11 ± 0.11 ^f^
Cellulose_m_	1 BT	32.04 ± 0.03 ^a^	21.23 ± 0.10 ^a^
	2 BT	23.04 ± 0.10 ^b^	18.21 ± 0.17 ^c^
	3 BT	18.81 ± 0.08 ^c^	15.00 ± 0.05 ^e^

Values within a column in the same group followed by the same letter are not significantly different (*p* > 0.05).

**Table 2 polymers-15-03376-t002:** Color values of cellulose from coconut coir with different bleaching times.

Samples	Bleaching Times	L*	a*	b*	WI	∆E
Cellulose_y_	1 BT	9.80 ± 0.08 ^f^	5.91 ± 0.03 ^c^	8.24 ± 0.09 ^b^	11.21 ± 0.05 ^d^	3.60 ± 0.05 ^d^
	2 BT	13.64 ± 0.06 ^b^	6.43 ± 0.04 ^a^	7.54 ± 0.08 ^d^	13.07± 0.04 ^c^	3.93 ± 0.04 ^c^
	3 BT	15.41 ± 3.00 ^ab^	6.49 ± 0.05 ^a^	8.15 ± 0.05 ^c^	15.80 ± 0.09 ^a^	1.87 ± 0.09 ^f^
Cellulose_m_	1 BT	11.86 ± 0.04 ^d^	6.22 ± 0.09 ^b^	5.66 ± 0.04 ^e^	9.23 ± 0.03 ^e^	9.80 ± 0.08 ^a^
	2 BT	12.34 ± 0.04 ^c^	6.84 ± 0.06 ^a^	10.85 ± 0.08 ^a^	11.40 ± 0.06 ^d^	6.51 ± 0.06 ^b^
	3 BT	16.23 ± 0.06 ^a^	6.23 ± 0.04 ^b^	8.72 ± 0.03 ^b^	14.77 ± 0.08 ^b^	2.26 ± 0.03 ^e^

Values within a column in the same group followed by the same letter are not significantly different (*p* > 0.05).

**Table 3 polymers-15-03376-t003:** Color values of CMC from coconut coir with different bleaching times.

Samples	Bleaching Times	L*	a*	b*	WI	∆E
CMC_y_	1 BT	42.17 ± 0.03 ^b^	9.55 ± 0.01 ^c^	5.63 ± 0.08 ^d^	41.11 ± 0.05 ^c^	13.06 ± 0.05 ^c^
	2 BT	42.62 ± 0.05 ^b^	9.97 ± 0.37 ^b^	5.56 ± 0.07 ^d^	41.49 ± 0.04 ^b^	15.89 ± 0.04 ^b^
	3 BT	81.81 ± 0.02 ^a^	11.08 ± 0.02 ^a^	11.20 ± 0.04 ^b^	75.35 ± 0.01 ^a^	39.60 ± 0.01 ^a^
CMC_m_	1 BT	7.44 ± 0.10 ^e^	5.14 ± 0.04 ^e^	9.63 ± 1.19 ^c^	16.72 ± 0.32 ^f^	2.75 ± 0.32 ^f^
	2 BT	8.45 ± 0.04 ^d^	6.29 ± 0.02 ^d^	9.95 ± 0.06 ^c^	17.60 ± 0.10 ^e^	3.84 ± 0.10 ^e^
	3 BT	10.81 ± 0.03 ^c^	5.57 ± 0.03 ^e^	13.64 ± 0.04 ^a^	29.25 ± 0.08 ^d^	12.91 ± 0.08 ^d^

Values within a column in the same group followed by the same letter are not significantly different (*p* > 0.05).

**Table 4 polymers-15-03376-t004:** Functional properties of cellulose_y_ and cellulose_m_.

Samples	Bleaching Times	Water Absorption Capacity (%)	Oil Absorption Capacity (%)	Water Holding Capacity (%)	Oil Holding Capacity (%)	Water Retention Capacity (%)	Oil Retention Capacity (%)	Water Solubility (%)	Moisture Content (%)
Cellulose_y_	1 BT	9.48 ± 0.07 ^f^	21.12 ± 0.01 ^c^	16.34 ± 0.43 ^d^	10.16 ± 0.04 ^c^	60.12 ± 0.02 ^c^	3.91 ± 0.02 ^b^	9.21 ± 0.01 ^e^	1.21 ± 0.05 ^f^
	2 BT	10.36 ± 0.45 ^e^	26.21 ± 0.16 ^b^	17.02 ± 0.02 ^b^	10.89 ± 0.02 ^b^	64.28 ± 0.10 ^b^	2.13 ± 0.02 ^f^	10.42 ± 0.02 ^b^	1.36 ± 0.05 ^e^
	3 BT	10.87 ± 0.02 ^c^	28.07 ± 0.02 ^a^	17.71 ± 0.20 ^a^	11.40 ± 0.35 ^a^	66.25 ± 0.05 ^a^	2.83 ± 0.08 ^d^	10.43 ± 0.02 ^b^	1.48 ± 0.05 ^d^
Cellulose_m_	1 BT	10.66 ± 0.23 ^d^	18.12 ± 0.01 ^f^	15.46 ± 0.34 ^f^	7.06 ± 0.03 ^f^	50.12 ± 0.02 ^f^	3.11 ± 0.02 ^c^	9.93 ± 0.01 ^d^	1.52 ± 0.02 ^c^
	2 BT	11.62 ± 0.26 ^b^	19.91 ± 0.07 ^e^	15.90 ± 0.01 ^e^	8.10 ± 0.06 ^e^	53.12 ± 0.01 ^e^	4.39 ± 0.16 ^a^	10.22 ± 0.01 ^c^	1.79 ± 0.07 ^b^
	3 BT	11.82 ± 0.14 ^a^	20.28 ± 0.11 ^d^	16.57 ± 0.23 ^c^	8.84 ± 0.09 ^d^	56.62 ± 0.01 ^d^	2.72 ± 0.13 ^e^	11.60 ± 0.01 ^a^	1.87 ± 0.06 ^a^

Values within a column in the same group followed by the same letter are not significantly different (*p* > 0.05).

**Table 5 polymers-15-03376-t005:** Functional properties of CMC_y_ and CMC_m_ with different bleaching times compared with commercial CMC.

Samples	Bleaching Times	Water Absorption Capacity (%)	Oil Absorption Capacity (%)	Water Holding Capacity (%)	Oil Holding Capacity (%)	Water Retention Capacity (%)	Oil Retention Capacity (%)	Water Solubility (%)	Moisture Content (%)
Commercial CMC	-	125.00 ± 0.03 ^a^	0.50 ± 0.05 ^e^	75.16 ± 0.04 ^a^	4.51 ± 0.02 ^e^	98.09 ± 0.07 ^a^	0.43 ± 0.01 ^e^	100.00 ± 0.02 ^a^	0.91± 0.02 ^g^
CMC_y_	1 BT	41.94 ± 0.07 ^d^	1.21 ± 0.02 ^d^	12.09 ± 0.02 ^d^	9.07 ± 0.02 ^d^	48.19 ± 0.07 ^e^	1.11 ± 0.01 ^d^	80.11 ± 0.01 ^f^	1.78 ± 0.02 ^a^
	2 BT	41.64 ± 0.24 ^d^	1.27 ± 0.03 ^d^	13.05 ± 0.01 ^b^	9.12 ± 0.02 ^d^	50.19 ± 0.12 ^d^	1.16 ± 0.01 ^c^	82.22 ± 0.01 ^e^	1.73 ± 0.03 ^b^
	3 BT	41.25 ± 0.02 ^d^	1.38 ± 0.01 ^c^	13.47 ± 0.40 ^b^	9.68 ± 0.07 ^b^	50.44 ± 0.05 ^d^	1.17 ± 0.02 ^c^	88.51 ± 0.34 ^c^	1.71 ± 0.05 ^c^
CMC_m_	1 BT	42.93 ± 0.05 ^b^	3.12 ± 0.02 ^b^	11.87 ± 0.03 ^e^	9.24 ± 0.12 ^c^	50.23 ± 0.10 ^d^	1.20 ± 0.01 ^b^	84.34 ± 0.01 ^d^	1.16 ± 0.02 ^d^
	2 BT	42.55 ± 0.11 ^c^	3.32 ± 0.01 ^b^	12.09 ± 0.01 ^d^	9.83 ± 0.02 ^b^	52.90 ± 0.01 ^c^	1.28 ± 0.02 ^b^	88.12 ± 0.01 ^c^	1.15 ± 0.07 ^e^
	3 BT	42.10 ± 0.01 ^c^	3.49 ± 0.01 ^a^	12.55 ± 0.15 ^c^	9.90 ± 0.08 ^a^	53.08 ± 0.06 ^b^	1.31 ± 0.04 ^a^	89.72 ± 0.01 ^b^	1.14 ± 0.03 ^f^

Values within a column in the same group followed by the same letter are not significantly different (*p* > 0.05).

## Data Availability

All the data are available within this manuscript.

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
