# Peer review of "Effect of Bleaching Processes on Physicochemical and Functional Properties of Cellulose and Carboxymethyl Cellulose from Young and Mature Coconut Coir"

_polymers, 2023, doi:10.3390/polym15163376_

Round 1

Reviewer 1 Report

Comments

My comments for the manuscript entitled, ‘Effect of Bleaching Processes on Physicochemical and Functional Properties of Cellulose and Carboxymethyl Cellulose 3 from Young and Mature Coconut Coir’, are given below,

1.      The authors used ‘coconut coir’ to extract cellulose and carboxymethyl cellulose, but in many places in the manuscript, the word ‘coconut powder’ is used instead. Check the uniformity of the raw material throughout the manuscript.

2.      In the experimental section, for the extraction of CMC from coconut coir, it is mentioned that 50 mL of NaOH is used, but its concentration is not mentioned.

3.      The FT-IR spectra of cellulose and CMC from young and matured coconut coir looks similar. There are no difference in the peaks between Cellulose and CMC (1 BT, 2 BT and 3 BT). NMR analysis should be included to confirm the structural differences.

4.      The work looks similar with the reported work (doi link )

Polymers 2021, 13, 81. https://dx.doi.org/10.3390/polym13010081

Polymers 2022, 14, 1872. https://doi.org/10.3390/polym14091872

Try to reduce the similarity between the two works, and highlight and describe the novelty of the present work.

5.      Manuscript should be thoroughly checked for language correction. Many grammatical errors are found, which should be reduced.

After modifying the manuscript according to the given comments, it can be accepted for publication.

Comments

My comments for the manuscript entitled, ‘Effect of Bleaching Processes on Physicochemical and Functional Properties of Cellulose and Carboxymethyl Cellulose 3 from Young and Mature Coconut Coir’, are given below,

1.      The authors used ‘coconut coir’ to extract cellulose and carboxymethyl cellulose, but in many places in the manuscript, the word ‘coconut powder’ is used instead. Check the uniformity of the raw material throughout the manuscript.

2.      In the experimental section, for the extraction of CMC from coconut coir, it is mentioned that 50 mL of NaOH is used, but its concentration is not mentioned.

3.      The FT-IR spectra of cellulose and CMC from young and matured coconut coir looks similar. There are no difference in the peaks between Cellulose and CMC (1 BT, 2 BT and 3 BT). NMR analysis should be included to confirm the structural differences.

4.      The work looks similar with the reported work (doi link )

Polymers 2021, 13, 81. https://dx.doi.org/10.3390/polym13010081

Polymers 2022, 14, 1872. https://doi.org/10.3390/polym14091872

Try to reduce the similarity between the two works, and highlight and describe the novelty of the present work.

5.      Manuscript should be thoroughly checked for language correction. Many grammatical errors are found, which should be reduced.

After modifying the manuscript according to the given comments, it can be accepted for publication.

Author Response

RESPONSES TO REFEREE #1’S COMMENTS:

  1. The authors used ‘coconut coir’ to extract cellulose and carboxymethyl cellulose, but in many places in the manuscript, the word ‘coconut powder’ is used instead. Check the uniformity of the raw material throughout the manuscript.

Answer: Please note the changes we have made throughout this manuscript.

  1. In the experimental section, for the extraction of CMC from coconut coir, it is mentioned that 50 mL of NaOH is used, but its concentration is not mentioned.

Answer: The concentration of NaOH had already been mentioned in line 144.

  1. The FT-IR spectra of cellulose and CMC from young and matured coconut coir looks similar. There is no difference in the peaks between Cellulose and CMC (1BT, 2BT and 3BT). NMR analysis should be included to confirm the structural differences.

Answer: This is an interesting point and one that made us think a great deal about our findings. However, we are not able to include the NMR analysis in this manuscript due to the limitation of revision time. However, we have added the results of the quantity analysis of XRD in section 3.5 and compared the functional properties of CMC from coconut coir with commercial CMC in Table 6.

  1. The work looks similar with the reported work (DOI link)

Polymers 2021, 13, 81. https://dx.doi.org/10.3390/polym13010081

Polymers 2022, 14, 1872. https://doi.org/10.3390/polym14091872

Try to reduce the similarity between the two works, and highlight and describe the novelty of the present work.

Answer: Thank you very much for your suggestion. The highlight and novelty of this work have been added to the abstract. have been reported. We have also reduced the similarity between the previous work and this work by adding the discussion on lignin content, IR spectra, crystallinity index, functional properties, DS value, and viscosity of both coconut coir.

  1. Manuscript should be thoroughly checked for language correction. Many grammatical errors are found, which should be reduced.

Answer: We apologize for grammatical errors. The manuscript has been edited for proper English language, grammar, punctuation, spelling, and overall style. We hope the revised manuscript will meet the requirements of academic publishing in Polymers.

Reviewer 2 Report

The experimental article “Effect of Bleaching Processes on Physicochemical and Functional Properties of Cellulose and Carboxymethyl Cellulose from Young and Mature Coconut Coir” is devoted to the isolation of cellulose from a poorly studied source of cellulose-containing raw materials of coconut coir for subsequent carboxymethylation by a suspension method. The positive aspects of the article include the scientific substantiation of the possibility of using waste to obtain cellulose and its chemical modification, as well as the use of modern methods for the study of cellulose. But the authors, who have experience in carboxymethylation of the purest known cellulose, namely, bacterial nanocellulose, do not disclose the reasons for the lack of publications on the production of cellulose from coconut coir, although they have information. The authors repeatedly applied the bleaching of cellulose and even CMC obtained from it in order to approach the required characteristics for food grade CMC. But for objective reasons, it is obvious that the course of research presented in the work does not allow this to be done. However, the authors claim that coconut coir CMC can be used in the food industry. Noting the positive aspects of this manuscript, I consider it necessary to comment on the fundamental reasons for the impossibility of obtaining pure cellulose from the indicated raw materials, the problems of carboxymethylation of cellulose with a content of 20-30% lignin, and also to compare the functional properties of CMC from coconut coir with commercial CMC.

Notes:

1. The abstract must reflect the novelty of the results obtained in the amount of one sentence. At the moment, much attention has been paid to the role of bleaching, which is justified in the isolation of cellulose and has no justification in the synthesis of CMC.

2. The introduction sufficiently reflects the role of CMC in the industry, lists well the cellulose-containing raw materials for CMC, but does not indicate the real reason for not using coconut coir for CMC.

3. Note that source 22 is about methylcellulose, not carboxymethylcellulose.

4. Align the listing of methods in the "Materials and Methods" section with the results in the Discussion of results.

5. Compare your data on low yields of CMC from slurry production with known data from commercial pulp or alternative pulp sources.

6. It is necessary to give an explanation for the low values of DS of CMC.

7. Four functional properties of cellulose and CMC (Water/oil absorption capacity, Water/oil holding capacity, Water/oil retention capacity, Water solubility) are described. Provide data on which property determines the scope of CMC use with reference to the literature, as well as what values ​​of these properties should be strived for when using alternative commercial sources of cellulose.

8. Pulp samples have very high lignin values. Consequently, during the carboxymethylation of the sample, in addition to the carboxymethylation of cellulose, the carboxymethylation of lignin will also take place. Accordingly, the values of the degree of substitution given by the authors are the sum for carboxymethylated cellulose and carboxymethylated lignin. The authors ignore this issue.

9. What is the purpose of bleaching CMC? Remove colored compounds of carboxymethylated lignin? Write about it. In this case, it becomes clear why coconut coir was not previously offered as a raw material for the production of CMC.

10. Why are the frequencies corresponding to lignin in cellulose and CMC not described in the results of IR spectroscopy?

11. The abstract contains the sentence “H2O2 bleaching can support delignification by reducing hemicellulose and lignin, as evidenced by FTIR showing a sharp peak at wave number 1219 cm-1.”, and there is no discussion of this provision in the text of the article.

12. Give the quantitative results of XRD, compare the reduction in the degree of crystallinity in your studies with carboxymethylation with other similar studies. Using classical ideas about the decrease in the crystallinity of cellulose during its carboxymethylation, describe the phenomena you observed.

13. Lines 408-409. If the CMC bleaching process removes the carboxymethylated lignin, which is amorphous, why does the degree of crystallinity of CMC decrease? Provide quantitative data.

14. From the description of the section “3.7. Degree of substitution (DS) of CMCy and CMCm with different bleaching times” it follows that the authors deny the fact of carboxymethylation of lignin during the carboxymethylation of a cellulose sample with a very high content of lignin. The authors are wrong. In this case, the authors should explain directly in the text the low DS of CMC values they obtained in the carboxymethylation of coconut coir with the expected values in the carboxymethylation of commercial cellulose (0.5 to 1.5 [11]) by the suspension method. It is possible to discuss with alternative sources of cellulose. What is the fundamental reason?

15. When discussing the viscosity of cellulose and CMC derived from it, the degree of polymerization of the cellulose should be mentioned.

16. The authors have conducted extensive research on the effect of bleaching on the properties of coconut coir CMC. As a result, CMC samples were obtained, the possibility of using them in the food industry still needs to be justified. The article does not clearly justify the possibility of their use, so the conclusions "High purity of CMCy and CMCm can be generally utilized in various texture improvements in food and pharmaceutical applications" must be excluded. Also from the abstract.

17. Check the correct formatting of links 7, 19.

Author Response

RESPONSES TO REFEREE #2’S COMMENTS:

The experimental article “Effect of Bleaching Processes on Physicochemical and Functional Properties of Cellulose and Carboxymethyl Cellulose from Young and Mature Coconut Coir” is devoted to the isolation of cellulose from a poorly studied source of cellulose-containing raw materials of coconut coir for subsequent carboxymethylation by a suspension method. The positive aspects of the article include the scientific substantiation of the possibility of using waste to obtain cellulose and its chemical modification, as well as the use of modern methods for the study of cellulose. But the authors, who have experience in carboxymethylation of the purest known cellulose, namely, bacterial nanocellulose, do not disclose the reasons for the lack of publications on the production of cellulose from coconut coir, although they have information. The authors repeatedly applied the bleaching of cellulose and even CMC obtained from it in order to approach the required characteristics for food grade CMC. But for objective reasons, it is obvious that the course of research presented in the work does not allow this to be done. However, the authors claim that coconut coir CMC can be used in the food industry. Noting the positive aspects of this manuscript, I consider it necessary to comment on the fundamental reasons for the impossibility of obtaining pure cellulose from the indicated raw materials, the problems of carboxymethylation of cellulose with a content of 20-30% lignin, and also to compare the functional properties of CMC from coconut coir with commercial CMC.

Answer: Thank you very much for your insightful comment. We are hopeful that you agree that this revision is better targeted toward Polymers. We are hopeful that our revised focus helps to improve your opinion of our work. The impossibility of obtaining pure cellulose from coconut coir has been added in lines 86-98. The problems of carboxymethylation of cellulose with a content of 20-30% lignin have been added in lines 303-305. Moreover, the functional properties of CMC from coconut coir have now been compared with commercial CMC in Table 6.

  1. The abstract must reflect the novelty of the results obtained in the amount of one sentence. At the moment, much attention has been paid to the role of bleaching, which is justified in the isolation of cellulose and has no justification in the synthesis of CMC.

Answer: Thank you for your suggestions. The novelty of the results has been added in the abstract in lines 23-25.

  1. The introduction sufficiently reflects the role of CMC in the industry, lists well the cellulose-containing raw materials for CMC, but does not indicate the real reason for not using coconut coir for CMC.

Answer: This is an interesting comment. The real reason for not using coconut coir for CMC was lists in lines 86-98.

  1. Note that source 22 is about methylcellulose, not carboxymethylcellulose.

Answer: Thank you very much for your correction. This source is deleted.

  1. Align the listing of methods in the "Materials and Methods" section with the results in the Discussion of results.

Answer: Done.

  1. Compare your data on low yields of CMC from slurry production with known data from commercial pulp or alternative pulp sources.

Answer: Thank you very much for your suggestions. The percent yield of CMC from coconut coir and corn husk has been mentioned in lines 280-281.

  1. It is necessary to give an explanation for the low values of DS of CMC.

Answer: The low DS value of CMC from coconut coir has been explained in lines 613-622.

  1. Four functional properties of cellulose and CMC (Water/oil absorption capacity, Water/oil holding capacity, Water/oil retention capacity, Water solubility) are described. Provide data on which property determines the scope of CMC use with reference to the literature, as well as what values ​​of these properties should be strived for when using alternative commercial sources of cellulose.

Answer: You raise an important question regarding the functional properties of both cellulose and CMC. The crucial functional properties of CMC are varied depending on the industries in which they will be applied. The scope of CMC used was mentioned in lines 734-738, 749-750, and 764-766.

  1. Pulp samples have very high lignin values. Consequently, during the carboxymethylation of the sample, in addition to the carboxymethylation of cellulose, the carboxymethylation of lignin will also take place. Accordingly, the values of the degree of substitution given by the authors are the sum for carboxymethylated cellulose and carboxymethylated lignin. The authors ignore this issue.

Answer: This was an important addition to the paper. We added the information about this issue in lines 613-622.

  1. What is the purpose of bleaching CMC? Remove colored compounds of carboxymethylated lignin? Write about it. In this case, it becomes clear why coconut coir was not previously offered as a raw material for the production of CMC.

Answer: Thank you very much for your suggestion. Please see the discussion in lines 338-339.

  1. Why are the frequencies corresponding to lignin in cellulose and CMC not described in the results of IR spectroscopy?

Answer: Thank you for your concern. We have added the frequencies corresponding to lignin in cellulose and CMC in lines 400-406.

  1. The abstract contains the sentence “H2O2 bleaching can support delignification by reducing hemicellulose and lignin, as evidenced by FTIR showing a sharp peak at wave number 1219 cm-1.”, and there is no discussion of this provision in the text of the article.

Answer: We have improved the discussion about IR spectroscopy in lines 400-406.

  1. Give the quantitative results of XRD, compare the reduction in the degree of crystallinity in your studies with carboxymethylation with other similar studies. Using classical ideas about the decrease in the crystallinity of cellulose during its carboxymethylation, describe the phenomena you observed.

Answer: We added the information about the quantitative results of XRD as crystallinity index (CI). The method of CI was also added in lines 198-204. We also have revised the discussion including the results of CI according to your suggestion in line with the discussion in lines 439-455. The table of CI as obtained by XRD measurements on coconut coir was also reported in Table 4.

  1. Lines 408-409. If the CMC bleaching process removes the carboxymethylated lignin, which is amorphous, why does the degree of crystallinity of CMC decrease? Provide quantitative data.

Answer: This is a great point and one that made us think a great deal about our findings. We have revised the discussion including the results of CI according to your suggestion in lines 439-455.

  1. From the description of the section “3.7. Degree of substitution (DS) of CMCy and CMCm with different bleaching times” it follows that the authors deny the fact of carboxymethylation of lignin during the carboxymethylation of a cellulose sample with a very high content of lignin. The authors are wrong. In this case, the authors should explain directly in the text the low DS of CMC values they obtained in the carboxymethylation of coconut coir with the expected values in the carboxymethylation of commercial cellulose (0.5 to 1.5 [11]) by the suspension method. It is possible to discuss with alternative sources of cellulose. What is the fundamental reason?

Answer: You raise a very valid point about the Degree of substitution (DS) of CMC. We tried to discuss the fundamental reason for the low DS of CMC from coconut coir in lines 613-622.

  1. When discussing the viscosity of cellulose and CMC derived from it, the degree of polymerization of the cellulose should be mentioned.

Answer: Thank you for your suggestion. This is a very helpful frame for our manuscript. We have already included the discussion of the degree of polymerization of cellulose in lines 794-799.

  1. The authors have conducted extensive research on the effect of bleaching on the properties of coconut coir CMC. As a result, CMC samples were obtained, the possibility of using them in the food industry still needs to be justified. The article does not clearly justify the possibility of their use, so the conclusions "High purity of CMCy and CMCm can be generally utilized in various texture improvements in food and pharmaceutical applications" must be excluded. Also from the abstract.

Answer: Thank you very much for your concern. We have deleted the mentioned sentence from the conclusion and abstract also.

  1. Check the correct formatting of links 7, 19.

Answer: Done.

Round 2

Reviewer 1 Report

Accept in present form

Accept in present form

Author Response

Thank you

Reviewer 2 Report

The authors did not disregard any comment of the reviewer, made changes to the text of the article, expanded the description and provided new data. But in a hurry, there were shortcomings that should be eliminated.

1. Reference “Viera, R.G.P.; Filho, G.R.; de Assunção, R.M.N.; S. Meireles, C.d.; Vieira, J.G.; de Oliveira, G.S. Synthesis and characterization of methylcellulose from sugar cane bagasse cellulose. Carbohydrate Polymers 2007, 67, 182-189, 8 doi:10.1016/j.carbpol.2006.05.007." devoted to methylcellulose rather than carboxymethylcellulose, the authors moved it from one place to another.

2. Reference “Huang, C.M.; Chia, P.X.; Lim, C.S.; Nai, J.Q.; Ding, D. Y.; Seow, P. B.; Wong, C. W.; Chan, E.W. Synthesis and characterization of carboxymethyl cellulose from various agricultural wastes. cellul. Chem. Technol 2017, 51, 665-672.” the number has changed, now it is "38", not "40".

Please pay attention to your statement about the low yield of SMS in Huang's work. You are not right. In Huang's work, in fact CMC yield is given in g obtained from 5 grams of cellulose, so the yield is above 100%. Fix it.

3. I ask the authors to clarify their answer to my question directly in the article: what characteristic absorption frequency in the IR spectrum of cellulose and SMS from it corresponds to lignin and its carboxymethylation product (Figure 3)?

In development: such a high content of lignin in cellulose will definitely be reflected in the IR spectrum.

Your answer about the frequency of 1219 cm-1 is highly debatable. I think that the references given by you in the description of IR spectroscopy (46, 11, 47) are erroneous.

4. The scope of the SMS received by the authors is not protected. These samples cannot be used in the food industry, especially in the pharmaceutical industry.

5. Links 46 and 48 match. Please check all links for compliance with the citation in the article!

Author Response

RESPONSES TO REFEREE #1’S COMMENTS:

The authors did not disregard any comment of the reviewer, made changes to the text of the article, expanded the description, and provided new data. But in a hurry, there were shortcomings that should be eliminated.

  1. Reference “Viera, R.G.P.; Filho, G.R.; de Assunção, R.M.N.; S. Meireles, C.d.; Vieira, J.G.; de Oliveira, G.S. Synthesis and characterization of methylcellulose from sugar cane bagasse cellulose. Carbohydrate Polymers 2007, 67, 182-189, 8 doi:1016/j.carbpol.2006.05.007." devoted to methylcellulose rather than carboxymethylcellulose, the authors moved it from one place to another.

Answer: We apologized for our mistake. This reference has been removed from the manuscript.

  1. Reference “Huang, C.M.; Chia, P.X.; Lim, C.S.; Nai, J.Q.; Ding, D. Y.; Seow, P. B.; Wong, C. W.; Chan, E.W. Synthesis and characterization of carboxymethyl cellulose from various agricultural wastes. cellul. Chem. Technol 2017, 51, 665-672.” the number has changed, now it is "38", not "40".

Answer: Thank you for your correction. We have made a change in line 280.

  1. Please pay attention to your statement about the low yield of SMS in Huang's work. You are not right. In Huang's work, in fact CMC yield is given in g obtained from 5 grams of cellulose, so the yield is above 100%. Fix it.

Answer: Done

  1. I ask the authors to clarify their answer to my question directly in the article: what characteristic absorption frequency in the IR spectrum of cellulose and SMS from it corresponds to lignin and its carboxymethylation product (Figure 3)?

Answer: We apologize for the confusion in our previous response. We have added the discussion on characteristics absorption of cellulose and CMC in lines 393-397, 404-406 and 415-417.

  1. In development: such a high content of lignin in cellulose will definitely be reflected in the IR spectrum.

Your answer about the frequency of 1219 cm-1 is highly debatable. I think that the references given by you in the description of IR spectroscopy (46, 11, 47) are erroneous.

Answer: We have carefully checked the frequency and revised it in line 407 and cited the relevant references in this section.

  1. The scope of the SMS received by the authors is not protected. These samples cannot be used in the food industry, especially in the pharmaceutical industry.

Answer: We have mentioned the sentences that samples cannot be used in food and pharmaceutical industries, it is prudent to conduct toxicological evaluations before using CMC from coconut coir in lines 862-864.

  1. Links 46 and 48 Please check all links for compliance with the citation in the article!

Answer: We apologize for the reference problems in the revised manuscript. We have checked the entire manuscript carefully for reference errors.
